# Influence of Atomization Nozzles and Spraying Intervals on Growth, Biomass Yield, and Nutrient Uptake of Butter-Head Lettuce under Aeroponics System

Mazhar H. Tunio [ID], Jianmin Gao *, Imran A. Lakhiar [ID], Kashif A. Solangi, Waqar A. Qureshi [ID], Sher A. Shaikh [ID] and Jiedong Chen

School of Agricultural Equipment Engineering, Jiangsu University, Zhenjiang 212013, China; mazharhussaintunio@sau.edu.pk (M.H.T.); 5103160321@stmail.ujs.edu.cn (I.A.L.); 5103180312@stmail.ujs.edu.cn (K.A.S.); waqar128ahmed@gmail.com (W.A.Q.); sashaikh@sau.edu.pk (S.A.S.); 2221816033@ujs.edu.cn (J.C.)
* Correspondence: gaojianminujs@163.com; Tel.: +86-136-5528-2069

**Abstract:** The atomized nutrient solution droplet sizes and spraying intervals can impact the chemical properties of the nutrient solution, biomass yield, root-to-shoot ratio and nutrient uptake of aeroponically cultivated plants. In this study, four different nozzles having droplet sizes N1 = 11.24, N2 = 26.35, N3 = 17.38 and N4 = 4.89 µm were selected and misted at three nutrient solution spraying intervals of 30, 45 and 60 min, with a 5 min spraying time. The measured parameters were power of hydrogen (pH) and electrical conductivity (EC) values of the nutrient solution, shoot and root growth, ratio of roots to shoots (fresh and dry), biomass yield and nutrient uptake. The results indicated that the N1 presented significantly lower changes in chemical properties than those of N2, N3 and N4, resulting in stable lateral root growth and increased biomass yield. Also, the root-to-shoot ratio significantly increased with increasing spraying interval using N1 and N4 nozzles. The N1 nozzle also revealed a significant effect on the phosphorous, potassium and magnesium uptake by the plants misted at proposed nutrient solution spraying intervals. However, the ultrasonic nozzle showed a nonsignificant effect on all measured parameters with respect to spraying intervals. In the last, this research experiment validates the applicability of air-assisted nozzle (N1) misting at a 30-min spraying interval and 5 min of spraying time for the cultivation of butter-head lettuce in aeroponic systems.

**Keywords:** soilless cultivation; food security; climate change; aeroponics; nutrients management; air-assisted atomizer; root characteristics





## 1. Introduction

Presently, the global population continues to grow and has crossed a milestone in human history. A study reported that there have been significant global changes observed over the last decade, and food security remains a major concern. Besides, the unpredictable climate changes have negatively affected the availability of natural resources, such as water, agricultural land and food production [1,2]. Through flooding, hurricane, storms and droughts have drastically reduced agriculture land [3,4]. Scientists predicted that adverse weather conditions and climate change will result in the deprivation of the large areas of arable land, rendering them unstable for farming [5,6]. Accordingly, climate change and water scarcity are considered among the most significant problems and are expected to worsen in the future and could create serious food security issues for feeding large populations [7–10]. Recently, increased interest has been directed towards plant production in closed facilities such as plant factories, vertical farms and indoor-growing modules [11–14]. Indoor planting can provide a healthy environment for growing food and is not affected by climatic conditions [14]. Indoor agriculture also uses advanced

agricultural technologies to reduce water consumption by providing precise irrigation and effective scheduling [15–17].

Innovative agricultural techniques are advisable to adopt soilless cultivation systems that are independent of climate conditions, soil fertility and soil-borne diseases and do not require large spaces and intensive work labor. These plant cultivation techniques are the most intensive and uses all available resources efficiently and maximize the crop yield compared to that of the traditional agriculture [15]. The use of soilless cultivation for improved production and aeroponic cultivation is the most appropriate modern horticulture soilless practice [15]. This system can be defined as a closed air and water/nutrient ecosystem that promotes swift growth of plants with virtually no water, soil or medium [18–24]. Aeroponics uses mist or nutrient solution as an alternative to water, which is an effective and competent method for plant growth. Aeroponics can save 95% of the water used in conventional agricultural practices and requires minimal space. This technique has been applied successfully for cultivation of leafy vegetables, root vegetables, aromatic herbs and medicinal plants, because the nutritional quality and properties, such as phenolic compound, flavonoid, antioxidant and vitamin contents, were higher in aeroponic-grown plants than in other soilless plants and soil-grown plants [25]. Moreover, the aeroponics system is still scientifically unclear, and several aspects of the system, such as the proper selection of atomizers (droplets), spraying time, spraying interval, root-zone temperature, humidity and best nutrient solution for different varieties of vegetables, have yet to be investigated [26–28]. In aeroponics systems, the nutrient solution is misted through atomizers, such as mechanical atomizers and piezoelectric ultrasonic foggers. Additionally, the mechanical atomizers are further categorized as air-based and airless (high and low pressures) atomizers. However, the ultrasonic foggers are categorized as high, medium and low frequency [29–31].

In aeroponics systems, plants receive nutrient droplets directly, which can affect the plant root environment, therefore, the main problem of the system is the size of the nutrient droplets, because relatively large droplets could reduce the oxygen available to the root system [22,27]. Moreover, a relatively low oxygen concentration could affect root respiration and nutrient absorption, and the amount of oxygen available to the root environment is a highly significant factor for plant growth [22]. Besides, the small water droplets could produce many root hairs, and roots may not form a lateral root system and thus cannot continue to grow. Furthermore, choosing an appropriate nutrient solution misting method is very important for plant growth, because the frequency, atomization time and atomization interval could affect the physical and chemical properties and quality of the nutrient solution. Electrical conductivity (EC) and pH are the most important parameters of the nutrient solution, and thus far, studies have reported that the atomizer (droplet size) can change the EC and pH values of the nutrient solution after dissolution, resulting in a reduction in yield biomass [21].

Lettuce is one of the most widely cultivated vegetable crops in the world and is eaten as a salad green. Lettuce has become the focus of many studies due to its high nutritional value, and contents of minerals, vitamins and folic acid, which play an important role in the human diet and nutrition [32–36]. Hence, for this study, we selected the lettuce as the test crop. The objective of this study was to determine the proper droplet size (atomizer) and nutrient solution spraying interval of the aeroponic system and the effects on the chemical properties of the nutrient solution, root-to-shoot ratio and nutrient uptake of aeroponically cultivated lettuce.

## 2. Materials and Methods

### 2.1. Experimental Site and Climate Conditions

The experiment was conducted in the semi-controlled Venlo-type greenhouse of Jiangsu University P.R. China (33.57° N, 118.16° E), during November–December 2019. For the determination of climate conditions, an automatic weather station (Hobo U12-012, Onset Computer Corp.) was placed in the center of the aeroponics systems. The measured

maximum and minimum average temperatures were 20.13 ± 1.32 °C and 7.65 ± 0.89 °C respectively, in the greenhouse, and 16.46 ± 0.72 °C and 4.17 ± 1.02 °C respectively, in the growth boxes of the aeroponics systems. Likewise, the maximum and minimum relative humidity were 68.43% ± 5.13% and 45.73% ± 2.73% respectively, in the greenhouse, and 89.4% ± 3.56% and 52.35% ± 1.98% respectively, in the growth chamber of the aeroponics systems.

### 2.2. Aeroponic System and Nozzles

For present study, the aeroponic system previously reported by our research team [37] was utilized. The aeroponic systems were manufactured with blue polyethylene plastic containers with a Styrofoam lid fixed in a steel frame. Additionally, the system was composed of four different atomizers (droplet sizes), a pressure pump (model PLD-1206, Prandy Electromechanical Equipment Co., Ltd., Shijiazhuang City, China), air compressors (model 750-30<2530>, Shengyuan Air Compressor Manufacturing Co., Ltd., Wenling, Zhenjiang, China), a mist pipeline, a reflux pipeline, a fluid infusion measuring pump, an axial flow fan, plastic cups to hold the plants, a nutrient solution tank, a demographic pump pressure regulator, a stainless-steel net and connectors (two- and three-way connectors). The schematic view of the aeroponics system is displayed in Figure 1.

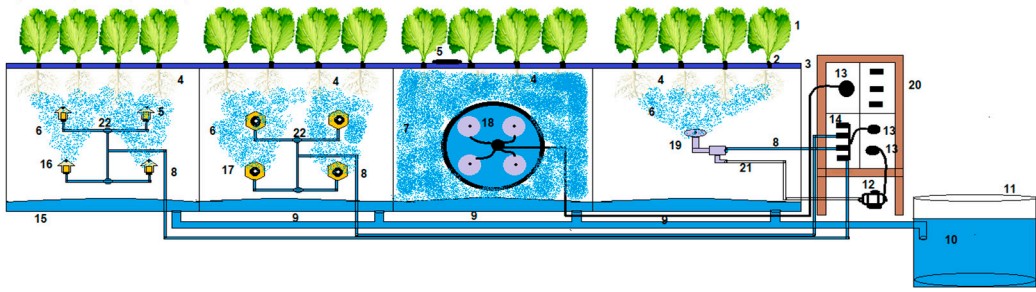

**Figure 1.** Layout of the experimental setup: 1, plants; 2, plant holder; 3, lid; 4, plant roots; 5, fog circulation fan; 6, nutrient solution mist; 7, nutrient solution fog; 8, nutrient misting line; 9, drainage line; 10, nutrient solution; 11, nutrient solution tank; 12, air compressor; 13, timers; 14, pressure pump; 15, growth boxes; 16 low-pressure nozzle; 17, low-pressure nozzle; 18, ultrasonic nozzles; 19, high-pressure nozzle with air; 20, electric box; 21, air inlet; 22, connectors.

### 2.3. Measurement of Droplet Sizes

The droplet size distribution is known as a very important parameter for the hydraulic performance of any nozzle/atomizer. A laser particle size analyzer (Winner318B, Jinan Winner Particle Instruments Stock Co., Ltd., Jinan, China), a computer, an air compressor (OTS-550, Taizhou Outstanding Industry and Trade Co., Ltd., Taizhou, China) and a pressure pump (PLD-1206, Shijiazhuang Pulandi Machine Equipment Co., Ltd., Wenling, Zhenjiang, China) were used to determine the particle size distribution, as shown in Figure 2. The instrument was composed of a photodiode detector, laser transmitter and storage circuit imaging system. More importantly, the working pressures of the air compressor, water pump and water flow rate for the air-assisted atomizer were 0.4 MPa, 0.2 MPa and 4 L min$^{-1}$, respectively. A pump pressure of 0.2 MPa and a water flow rate of 1 L min$^{-1}$ were stable for each airless atomizer, and the flow rate of each ultrasonic fogger was 1 L min$^{-1}$. More importantly, the flow rates and pressures were kept constant for the air compressor and water pump throughout the cultivation period.

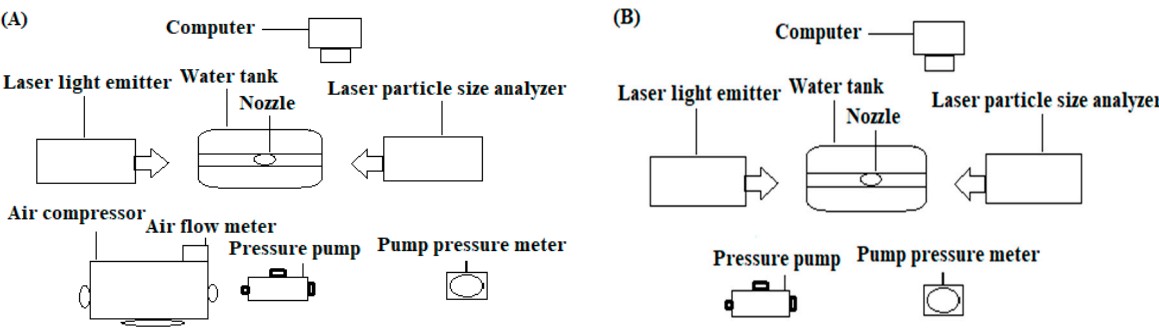

**Figure 2.** Layout of experimental setup for the measurement of droplet size: (**A**) high-pressure nozzles (with air), (**B**) low-pressure nozzles (without air).

### 2.4. Plant Material and Experimental Arrangement

Butter-head lettuce seeds were cultivated in expanded polystyrene (EPS) trays with 72 cells containing an equal quantity of perlite material. Emerged lettuce seedlings were watered four times a week for two weeks. On the sixteenth day of cultivation, the lettuce seedlings were transplanted into the aeroponics systems. The aeroponics systems were arranged in a randomized complete block design (RCBD) with 12 treatments. The experiment was comprised of four atomizing nozzles (one with air, two without air and one ultrasonic nozzle) and three nutrient solution spraying intervals (30, 45 and 60 min). Moreover, the systems spraying time of 5 min was constant throughout the experiment. The details of the experimental treatments are: nozzles with air (N1) implemented with 30-, 45- and 60-min spraying intervals (I) denoted as N1I1, N1I2 and N1I3, respectively. The nozzles without air (N2 and N3) operated at the same three spraying intervals and were denoted as N2I1, N2I2 and N2I3, and N3I1, N3I2 and N3I3, respectively. The ultrasonic nozzle (N4) misting at 30-, 45- and 60-min spraying intervals was denoted as N4I1, N4I2 and N4I3, respectively. Each aeroponic system consisted of 12 lettuce seedlings and was replicated three times, and there were 432 plants in total. The plant-to-plant distance was maintained at $14 \times 16$ cm$^2$ in each aeroponic system. The Hoagland's chemical composition of nutrient solution with 945 mg L$^{-1}$ of Ca (NO$_3$)$_2 \bullet 4$H$_2$O, 607 mg L$^{-1}$ of KNO$_3$, 115 mg L$^{-1}$ of NH$_4$H$_2$PO$_4$, 493 mg L$^{-1}$ of MgSO$_4 \bullet 7$H$_2$O, 2.86 mg L$^{-1}$ of H$_3$BO$_3$, 2.13 mg L$^{-1}$ of MnSO$_4 \bullet 4$H$_2$O, 0.22 mg L$^{-1}$ of ZnSO$_4 \bullet 7$H$_2$O, 0.08 mg L$^{-1}$ of CuSO$_4 \bullet 5$H$_2$O, 0.02 mg L$^{-1}$ of (NH$_4$)$_6$Mo$_7$O$_{24} \bullet 4$H$_2$O and 28 mg L$^{-1}$ of ferrum-Ethylenediaminetetraacetic Acid (Fe-EDTA) was misted though atomizers at the proposed spraying intervals throughout the experiment. A fresh nutrient solution with pH (5.8–6) and EC (1.6–2.2 dS m$^{-1}$) values was replaced with recycled nutrient solution of each system on the fifth day throughout the experiment.

### 2.5. Measurement of the Power of Hydrogen (pH) and Electrical Conductivity (EC)

The EC and pH values of the atomized nutrient solution were measured in each treatment group every day. On every fifth day, a fresh nutrient solution was replaced with new nutrient solution. A conductivity meter (ProfiLine Cond 3110, WTW, Weilheim, Germany) with accuracies of 0.001 and 0.1 mS cm$^{-1}$ was used to measure the EC value of the Hoagland's nutrient solution. For the determination of the pH value of the nutrient solutions of all treatments, a pH meter (ProfiLine pH 3110, WTW, Weilheim, Germany) with accuracy of 0.01 was used.

### 2.6. Vegetative Growth Parameters of Lettuce Plants

To compare the vegetative growth parameters of 40 days after transplant (DAT) under all treatments, the number of leaves (NL), stem diameter (SD, mm), leaf length (LL, cm), leaf width (LW, cm), leaf area (LA, cm$^2$), total biomass yield (TBY, g plant$^{-1}$) and edible yield (EY, g plant$^{-1}$) were measured using previously reported procedures in References [38–41].

The NL was measured manually by counting the leaves per plant. A digital Vernier caliper was used to calculate the SD (mm). The LL (cm) and LW (cm) of lettuce plants were measured using a steel measuring scale. Additionally, a laser leaf area meter (CI-203, CID Bio-Science, Inc. USA) was used to calculate the LA (cm$^2$). TBY (stem + leaves + root) in grams and EY (stem + leaves) in grams were measured by an electronic analytical balance with an accuracy of 0.1 mg. The fresh weighed samples were inserted carefully into transparent paper envelopes and oven-dried at 105 °C for 24 h to determine dry weight. Figure 3 presents the shoot growth of butter-head lettuce under all treatments.

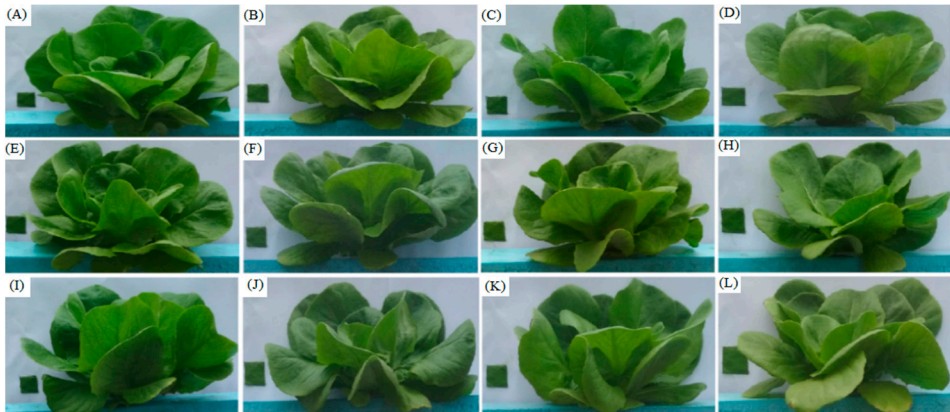

**Figure 3.** Shoot development of butter-head lettuce in aeroponics systems, (**A**) N1I1, (**B**) N2I1, (**C**) N3I1, (**D**) N4I1, (**E**) N1I2, (**F**) N2I2, (**G**) N3I2, (**H**) N4I2, (**I**) N1I3, (**J**) N2I3, (**K**) N3I3 and (**L**) N4I3 treatments.

*2.7. Root Characteristic Analysis*

For the analysis of average root diameter (cm), root length (cm), root area (cm$^2$), root volume (cm$^3$), maximum number of roots, median number of roots, network perimeter (cm) and root fresh weight (g plant$^{-1}$) of four plants from each treatment group of 40 day after transplant (DAT) were randomly selected for the measurement of root characteristics. The roots were washed and cleaned with paper towels. The roots were placed into a blue plastic container and photographed carefully from above with a digital camera. The photographs were scaled and treated with GiA Roots software (Georgia Tech Research Corporation and Duke University, USA) to measure the root characteristics of all treatments. The same procedure as shoot fresh weight was used to measure the root fresh weight. The root growth pattern of lettuce plants during the experiment are depicted in Figure 4.

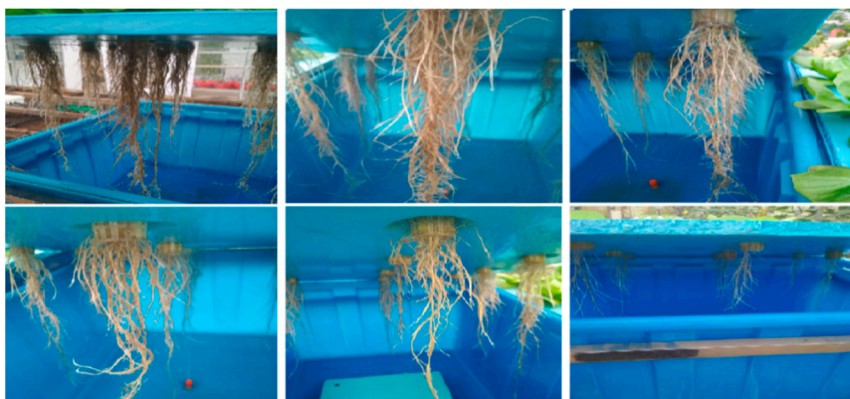

**Figure 4.** Root growth pattern under different treatments.

### 2.8. Nutrient Uptake

Four plants of lettuce from each treatment were randomly selected and used to determine the nutrient uptake. The total nitrogen (N) concentration was determined by using a Kjeldahl digestion method. The total phosphorous (P) concentration (molybdovanadate method) was measured by automated colorimetry, and the total potassium (K) concentration was determined by a flame photometric method [42]. Finally, calcium (Ca) and magnesium (Mg) were analyzed by using an optical emission spectrometer (Optima 5300 DV Spectrometer, Shelton, CT, USA) [33].

### 2.9. Statistical Analysis

SPSS Statistics 19.0 and Microsoft Excel 2016 software were used to analyze the data. The analysis of variance (ANOVA) was employed to determine the effect of different droplet sizes (nozzles) and nutrient solution spraying intervals on aeroponically grown butter-head lettuce. The correlation of the proposed parameters was tested by regression analyses and Duncan's multiple tests at the $p < 0.05$ level of significance.

## 3. Results

### 3.1. Droplet Size Measurement

The droplet size measurement of nozzles with and without air and ultrasonic nozzles operated at the same operational pressure is presented in Figure 5. The droplet sizes were calculated at frequencies of 10%, 25%, 50%, 75% and 90%. The results revealed that N2 had maximum droplet size, while the minimum and median droplet sizes were exhibited by N4 and N1, respectively. The average droplet size diameter (Dav.) for N1, N2, N3 and N4 was 11.24, 26.35, 17.38 and 4.89 µm, respectively.

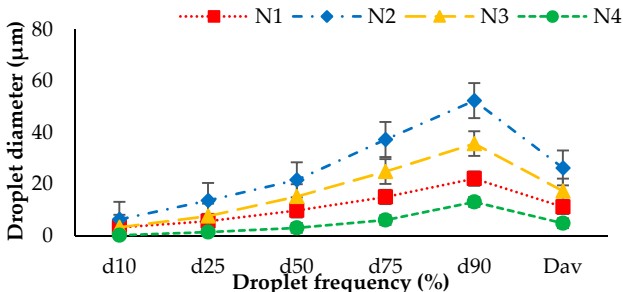

**Figure 5.** Comparison of droplet sizes (µm) distribution of nozzles with air (N1), without air (N2 and N3) and ultrasonic nozzles (N4).

### 3.2. Effect of Droplet Sizes and Spraying Intervals on the pH and EC Values of Hoagland's Nutrient Solution

The statistical analyses results for the effect of different droplet sizes (nozzles) and nutrient solution spraying intervals on pH and EC values of atomized nutrient solution are depicted in Figure 6. It was observed that the interaction between droplet size and spraying interval had a significant ($p < 0.05$) effect on the average values of pH and EC throughout the experiment. An increasing trend for pH under N1, N2 and N3, and a decreasing trend for EC, were observed for all four nozzles misted at 30-, 45- and 60-min nutrition solution spraying intervals. However, pH values showed a decreasing trend under the N4 atomizer at 30, 45 and 60 min of spraying interval. The results revealed that the highest and lowest change in pH values were 0.82 and 0.33 for N1, 0.88 and 0.49 for N2, 1.01 and 0.47 for N3 and 1.15 and 0.73 for N4 respectively, at 30-, 45- and 60-min nutrient solution spraying intervals, respectively. It was observed that the EC values were inversely proportional to the spraying interval for all nozzles. The minimum change in EC values was lowest for the 30-min interval and reached a maximum at the 60-min spraying interval for all tested nozzles (N1, N2, N3 and N4). The highest reduction in EC value (0.50 dS m$^{-1}$) was calculated for the ultrasonic nozzle (N4) misting at 60-min nutrient solution

spraying intervals, while the lowest change in EC values of 0.23 dS m$^{-1}$ was observed for the nozzle with air (N1) misting at 30-min spraying intervals. The ANOVA results indicated that nozzles with and without air showed a highly significant effect on the pH value at all evaluated intervals ($p < 0.001$), and the ultrasonic nozzle presented a nonsignificant ($p > 0.05$) effect. Accordingly, the interaction of nozzles and spraying intervals indicated a significant effect ($p < 0.05$) on EC values of the butter-head lettuce variety.

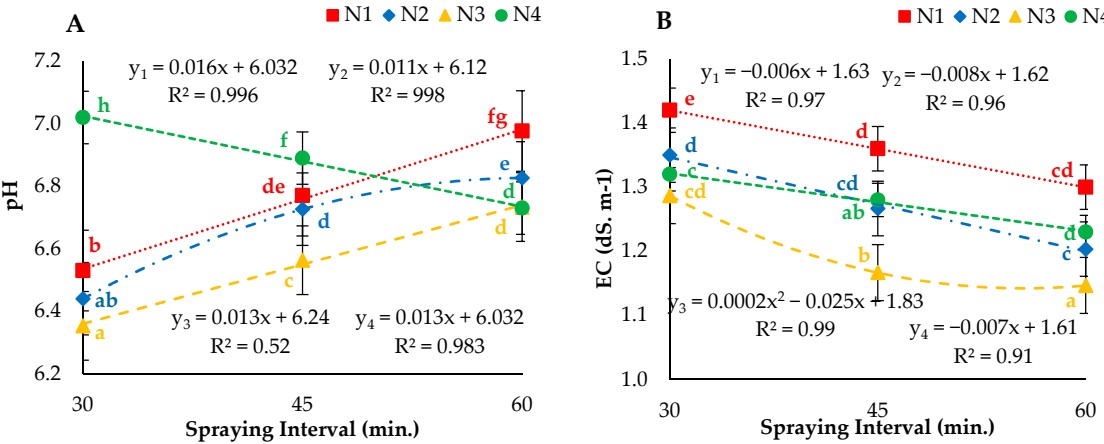

**Figure 6.** Average (**A**) change in pH and (**B**) change in electrical conductivity (EC) values of Hoagland's nutrient solution for N1, N2, N3 and N4 nozzles misting at 0 (initial), 30-, 45- and 60-min intervals. Whereas $y_n = N_n$.

### 3.3. Vegetative Growth Parameters

#### 3.3.1. Shoot Growth Parameters

The two-way ANOVA analyses results of NL, SD, LL, LW and LA are represented in Figure 7. The statistical analysis results showed that the interaction between droplet size and spraying interval had a significant ($p < 0.05$) effect on vegetative growth parameters. The results indicated that the average highest number of leaves per plant (28.7) was measured for N1I1, and the lowest (13.0) number of leaves per plant was observed for N4I3. Moreover, the maximum and minimum SDs of 3.84 and 0.98 mm were observed under the N1I2 and N4I1 treatments, respectively. Additionally, the droplet sizes and spraying intervals had positive significant ($p < 0.05$) effects on the leaf length, leaf width and leaf area of aeroponically grown butter-head lettuce. The maximum LL, LW and LA of 17.15 cm, 11.64 cm and 137.38 cm$^2$ respectively, were calculated under N1I3 treatment. However, the minimum LL, LW and LA of 6.28 cm, 2.75 cm and 9.65 cm$^2$ respectively, were observed for the ultrasonic nozzle operating with a 60-min spraying interval. More importantly, the regression analysis results revealed that N1 showed a significant ($p < 0.05$) effect on NL, SD, LW, LL and LA. Moreover, N2 presented a significant ($p < 0.05$) effect on LW and LA; additionally, N3 exhibited a significant ($p < 0.05$) effect on LL and LA. The interaction of the ultrasonic nozzle and spraying interval was nonsignificant ($p > 0.05$) for all proposed parameters.

#### 3.3.2. Correlation between Shoot Growth Parameter208B

The analyzed results of the correlation between growth parameters are presented in Table 1. The results showed that the growth parameters had strong positive correlations with each other under all treatments.

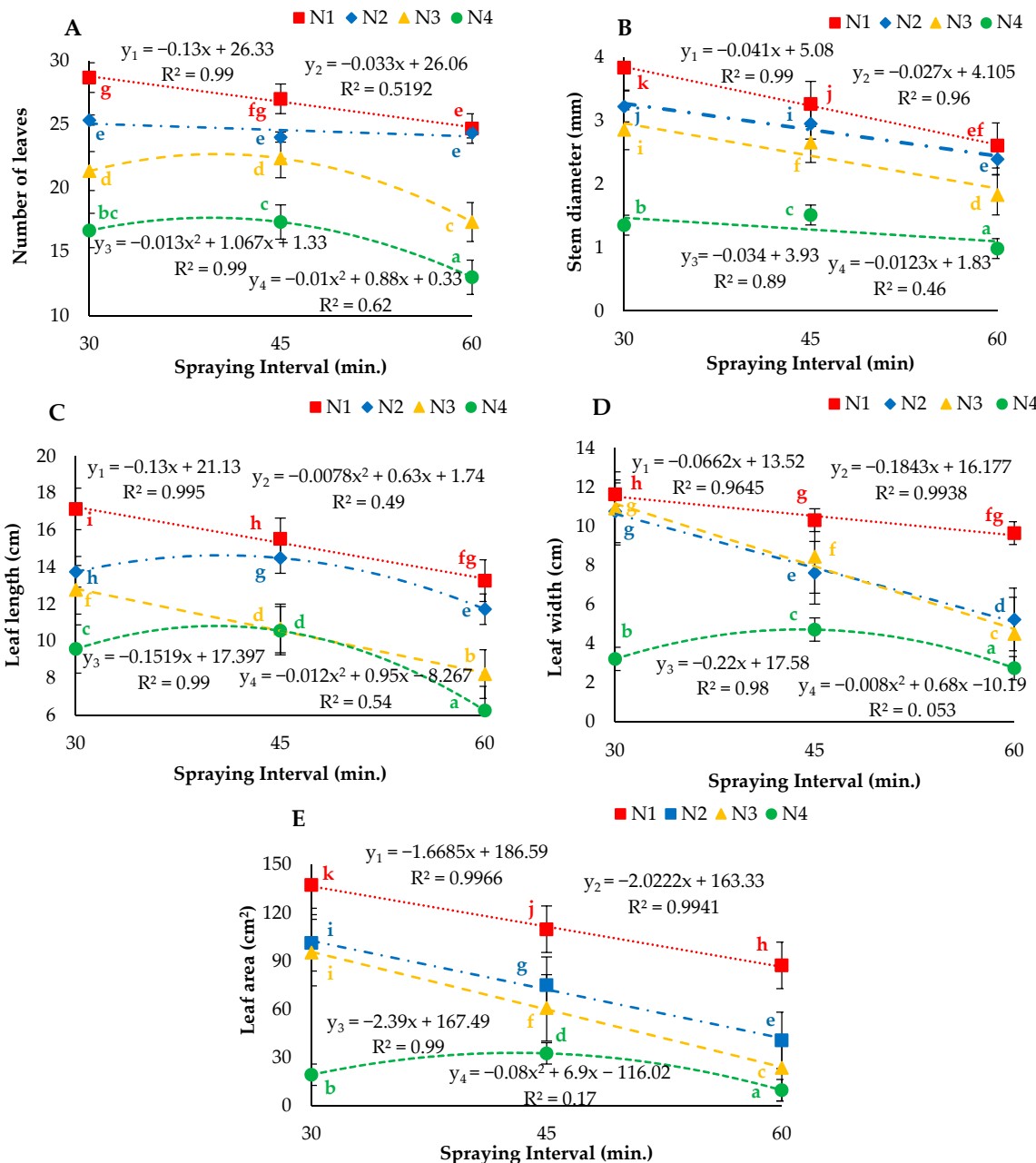

**Figure 7.** (**A**) Number of leaves, (**B**) stem diameter, (**C**) leaf length, (**D**) leaf width and (**E**) leaf area under different droplet sizes (N1, N2, N3 and N4) misted at 30-, 45- and 60-min intervals. Whereas $y_n = N_n$.

**Table 1.** Correlation analyses results between number of leaves (NL), shoot diameter (SD), leaf length (LL), leaf width (LW) and leaf area (LA).

| Parameters | NL | SD | LL | LW |
|---|---|---|---|---|
| SD | 0.95 | | | |
| LL | 0.94 | 0.93 | | |
| LW | 0.85 | 0.93 | 0.85 | |
| LA | 0.89 | 0.95 | 0.93 | 0.97 |

### 3.3.3. Total Biomass Yield and Edible Yield

The statistical analysis results of total biomass yield and edible yield are depicted in Figure 8. The results indicated that the interaction of droplet sizes and spraying intervals

had a significant ($p < 0.05$) effect on total biomass yield and edible yield. The highest and lowest total biomass yields of 63.85 and 9.29 g plant$^{-1}$ were measured for the N1I1 and N4I3 treatments, respectively. Furthermore, maximum and minimum edible yields of 49.48 and 4.92 g plant$^{-1}$ were observed for the N1I2 and N4I4 treatments, respectively. These findings indicated that the nozzles utilizing air showed a significant ($p < 0.05$) effect on the total biomass yield and edible yield of butter-head lettuce. Only the ultrasonic nozzle showed a very highly significant ($p \leq 0.01$) effect on total biomass yield. Moreover, the nozzles without air indicated a nonsignificant ($p > 0.05$) effect on both yield parameters when misting at 30-, 45- and 60-min intervals.

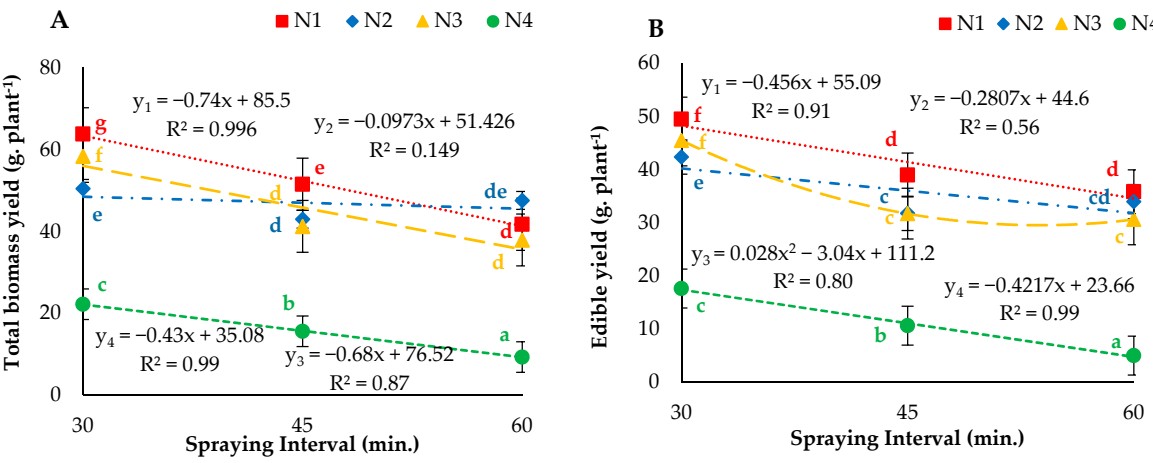

**Figure 8.** (**A**) Total biomass yield and (**B**) edible yield under different droplet sizes (N1, N2, N3 and N4) misted at 30-, 45- and 60-min intervals. Whereas $y_n = N_n$.

### 3.4. Root Characteristics

The droplet sizes (atomizers) and spraying intervals showed significant ($p < 0.05$) effects on the average root diameter, root length, root area, root volume, maximum number of roots, median number of roots, network perimeter and fresh weight of lettuce roots grown under aeroponic systems (Figure 9). A decreasing trend was observed under all treatments with respect to increasing spraying interval. The highest average root diameter (0.45 cm) and average root length (4707.8 cm) were observed under N1I1, and the lowest values of 0.34 and 2147 cm respectively, were calculated under the N4I3 treatment. Additionally, the maximum average root area, root volume, maximum number of roots, median number of roots and network perimeter of 1492.04 cm$^2$, 65.55 cm$^3$, 93.9, 47.5 and 8383.8 cm respectively, were observed under the N1I1 treatment. More importantly, the maximum and median numbers of roots for N1 misting at a 30-min interval were two to three times greater than those of N2, N3 and N4 misting at 45- and 60-min intervals. In general, the experimental results of the study indicated that roots could grow better using a nozzle implementing air and misting the nutrient solution at 30- and 45-min intervals. Accordingly, the nozzle with air (N1) showed a significant ($p < 0.05$) effect, N2 showed a nonsignificant ($p > 0.05$) effect and N3 and N4 presented a mixed phenomenon of significant ($p < 0.05$) and nonsignificant ($p > 0.05$) effects on all measured parameters when misting at 30-, 45- and 60-min intervals. From the regression analysis results of growth parameters, prediction models were developed.

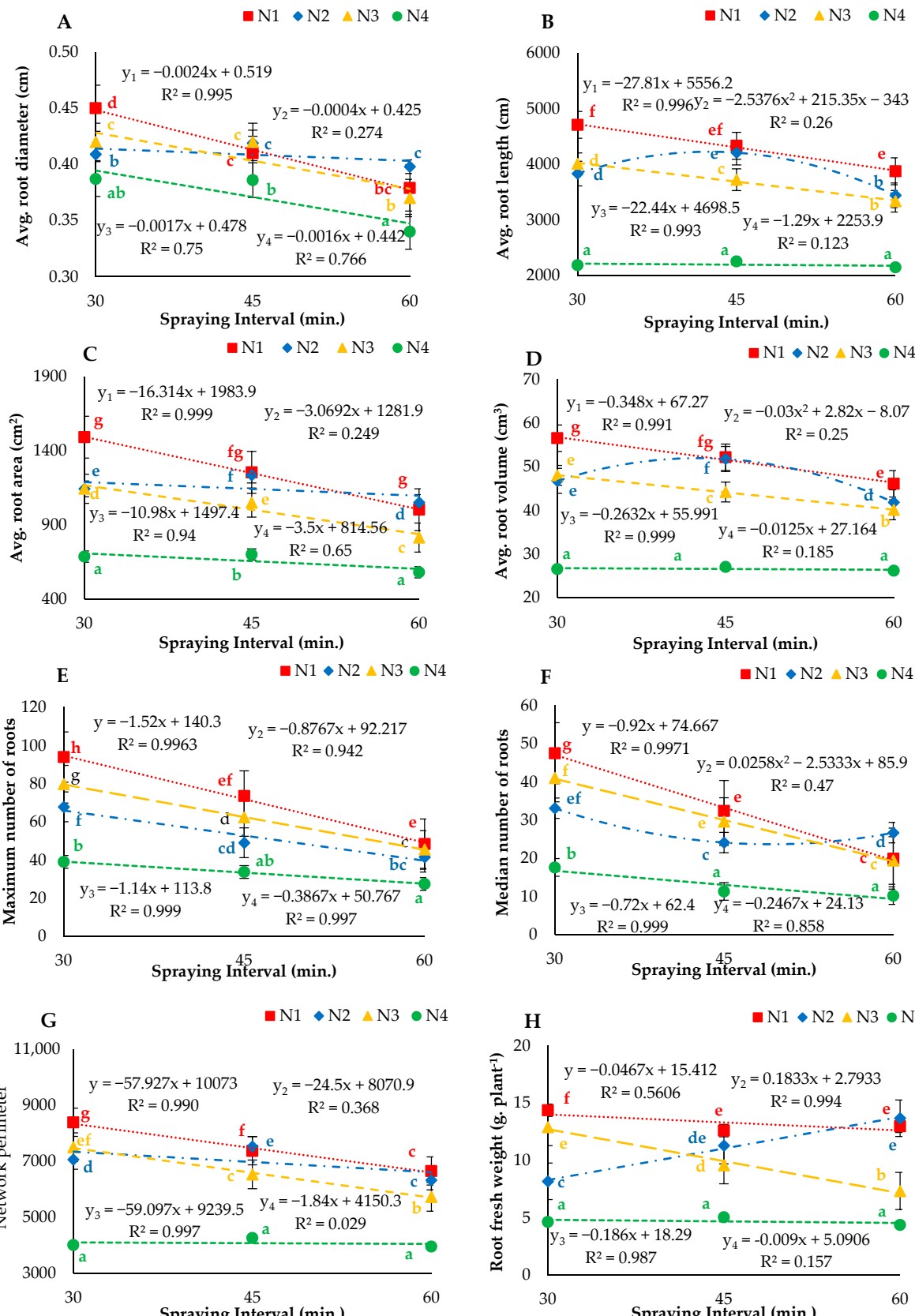

**Figure 9.** (**A**) Average root diameter, (**B**) average root length, (**C**) average root area, (**D**) average root volume, (**E**) maximum number of roots, (**F**) median number of roots (**G**) network perimeter, and (**H**) root fresh weight for different droplet sizes (N1, N2, N3 and N4) misted at 30-, 45- and 60-min intervals. Whereas $y_n = N_n$.

Correlation between Root Characteristics

The correlation between root growths parameters are presented in Table 2. The analyzed results showed a positive correlation between each other under all treatments.

**Table 2.** Correlation between root growth parameters.

|  | RD | RL | RA | RV | Max. Roots | Med. Roots | Net. Perimeter |
|---|---|---|---|---|---|---|---|
| RL | 0.77 | | | | | | |
| RA | 0.88 | 0.96 | | | | | |
| RV | 0.77 | 1.00 | 0.96 | | | | |
| Max. roots | 0.77 | 0.80 | 0.85 | 0.80 | | | |
| Med. roots | 0.82 | 0.78 | 0.83 | 0.77 | 0.94 | | |
| Net. perimeter | 0.81 | 0.99 | 0.97 | 0.99 | 0.83 | 0.81 | |
| RFW | 0.66 | 0.88 | 0.86 | 0.87 | 0.67 | 0.69 | 0.88 |

*3.5. Ratio of Roots to Shoots*

The analyzed results for the interaction effect of droplet size (nozzles) and spraying interval on the ratio of roots to shoots (fresh and dry) are displayed in Figure 10. The results showed that both parameters had a positive effect on the ratio of roots to shoots. The weight of shoot biomass was higher than that of roots under all treatments. The increasing root-to-shoot ratio for N1, N2 and N3 was observed when utilizing fresh weights of roots and shoots with respect to spraying interval, except for N4. Furthermore, a mixed trend of increasing and decreasing ratios was observed when utilizing dry weights. The regression analysis results indicated that N1 and N4 had a significant ($p < 0.05$) effect on the root-to-shoot ratio (fresh) and that N1 had a significant ($p < 0.05$) effect on the root-to-shoot ratio (dry). Additionally, N2 and N3 showed nonsignificant ($p > 0.05$) effects on the root-to-shoot ratios calculated using fresh and dry weights.

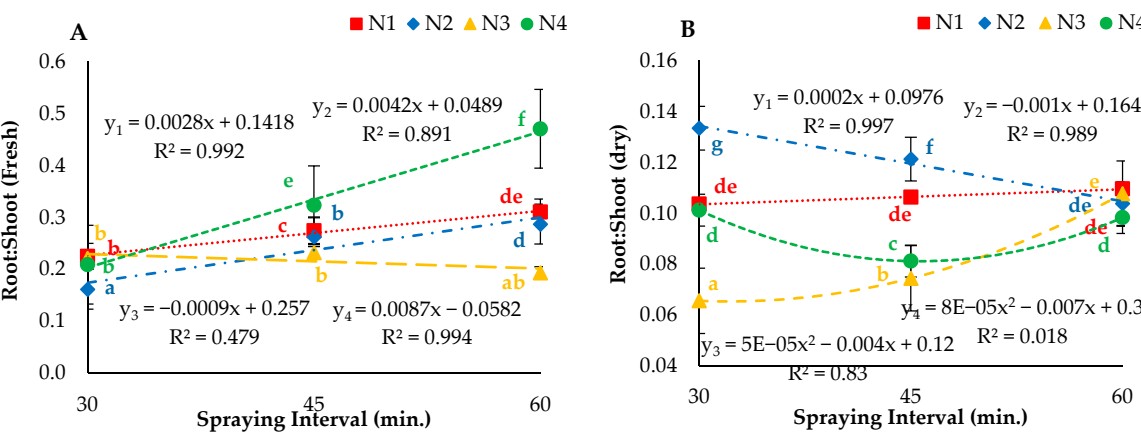

**Figure 10.** (**A**) Root: Shoot (fresh), (**B**) Root: Shoot (dry) under different droplet sizes (N1, N2, N3 and N4) misted at 30-, 45- and 60-min intervals. Whereas $y_n = N_n$.

*3.6. Nutrient Uptake*

The experimental results presented in Figure 11 describe the effect of different nozzles (droplet sizes) and nutrient solution spraying intervals on nutrient uptake. The regression analysis results revealed that the different atomizers operated at the three nutrient solution spraying intervals showed a significant ($p < 0.05$) effect on the nitrogen (N) uptake of lettuce leaves. The interaction of droplet size and spraying interval affected plant metabolism. A significant increase in N in leaf tissues increases photosynthesis efficiency and could result in high yield. The highest N uptake (0.36 mg. $g^{-1}$) was observed under the N1I1 treatment.

The results also revealed that the atomizers operating at 30- and 45-min spraying intervals had significantly higher values of N uptake compared to that at the 60-min spraying interval (Figure 11A). Furthermore, it was observed that both parameters affected P. The droplet size and spraying interval also affected plant metabolism and helped to simulate outdoor environmental conditions. The decrease in P was observed under N1, N2, N3 and N4 at 30-, 45- and 60-min nutrient solution spraying intervals. Moreover, different atomizers operating with a 30-min interval had significantly higher values than those of nozzles operating at 45- and 60-min intervals. Furthermore, the regression analysis results for the effects of different aeroponic nutrient solution spraying intervals on K uptake of the lettuce plants are displayed in Figure 11C. The highest uptake (0.60 mg. $g^{-1}$) was illustrated under the N1I1 treatment, and the lowest uptake (0.20 g. $plant^{-1}$) was calculated under N4I1. Mixed fractions of calcium (Ca) and magnesium (Mg) were observed under all treatments (Figure 11D,E). More importantly, the ultrasonic foggers (N4) operating at 30-, 45- and 60-min nutrient solution spraying intervals showed lower Ca and Mg concentrations than those observed for N1, N2 and N3.

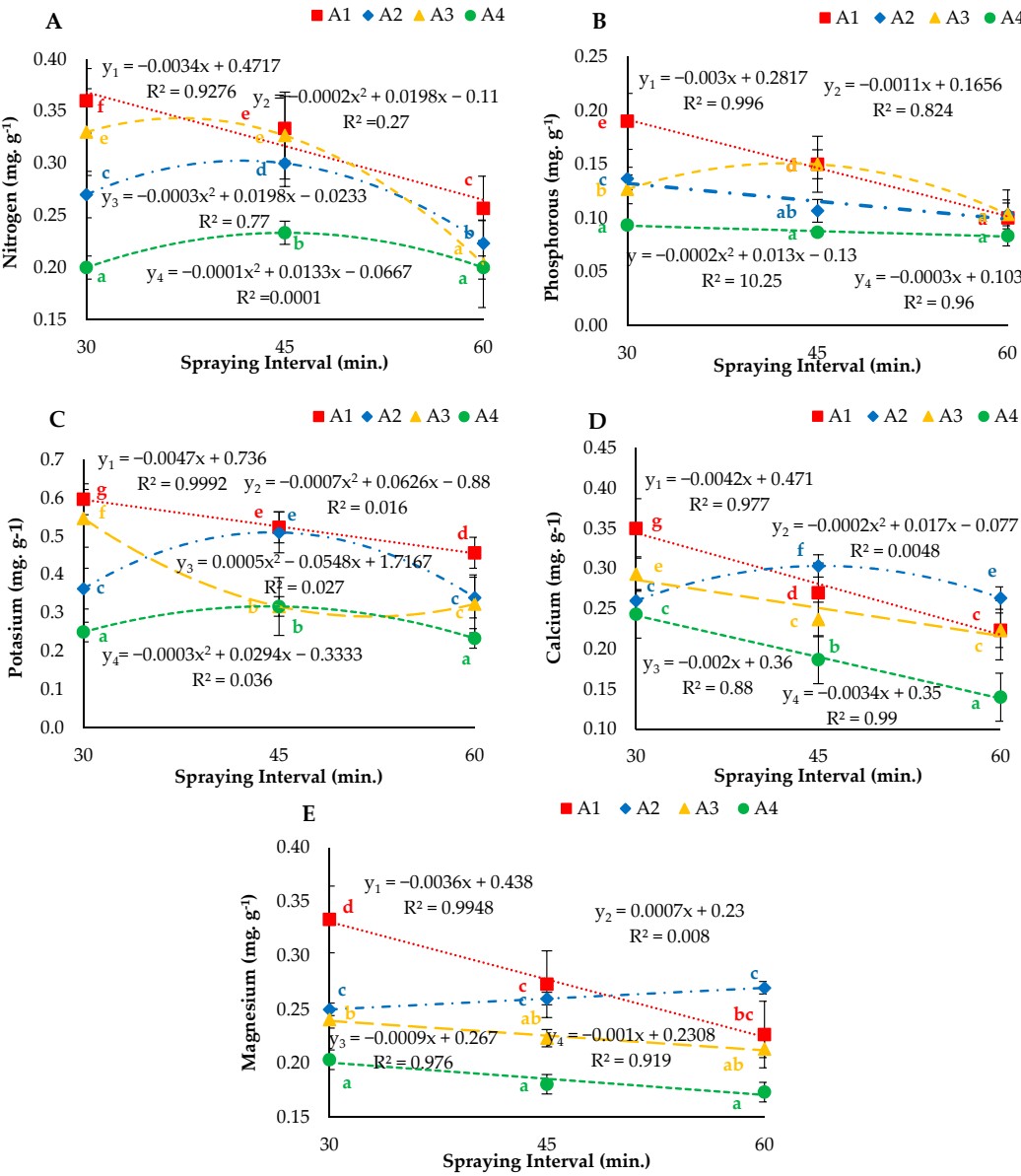

**Figure 11.** (**A**) Nitrogen, (**B**) phosphorus, (**C**) potassium, (**D**) calcium and (**E**) magnesium uptake under different droplet sizes (N1, N2, N3 and N4) of nutrient solution misted at 30-, 45- and 60-min intervals. Whereas $y_n = N_n$.

Correlation between Nutrient Uptakes

The analyzed results of the correlation between N, P, K, Ca and Mg uptake are depicted in Table 3. The results revealed that the measured parameters had moderate correlations with each other for the four different nozzles and the three different nutrient solution spraying intervals.

**Table 3.** Correlation between N, P, K, Mg and Ca.

| Parameters | N | K | P | Mg |
|:---:|:---:|:---:|:---:|:---:|
| K | 0.83 | | | |
| P | 0.86 | 0.64 | | |
| Mg | 0.78 | 0.84 | 0.59 | |
| Ca | 0.60 | 0.81 | 0.56 | 0.75 |

## 4. Discussion

Based on the literature, very few researchers have conducted studies on droplet size and spraying interval effects on the chemical properties of nutrient solutions, the growth of shoots to roots, the root-to-shoot ratio, the biomass yield and the nutrient uptake of lettuce crops grown in aeroponic systems. From the results, it was observed that droplet size (nozzles) had a significant ($p < 0.05$) effect on the chemical properties (EC and pH) of Hoagland's nutrient solution, except for the ultrasonic fogger, which showed a nonsignificant ($p > 0.05$) effect on pH value. Simultaneously, the pH values decreased, and EC values increased for all nozzles with respect to increasing spraying interval. The highest change in pH and change in EC were observed for the ultrasonic nozzle, and the lowest change in pH and EC values were calculated for the nozzle utilizing air. The findings of our research are in line with those of the authors of References [30,43], who concluded that nozzles (droplet size) could affect the physicochemical properties of nutrient solution and could change with time. In aeroponic systems, the reuse intervals of the nutrient solution vary from days to weeks. Similar results were also reported by the authors of Reference [21], who found that it is very important to evaluate the nutrient solution pH and EC of the nutrient solution regularly to avoid placing unnecessary pressure on the growth of plants. Researchers also reported that EC values lower than a certain range could reduce the availability of nutrients to plants, and EC values higher than this range could cause ion toxicity, unnecessary stress and an imbalance of nutrient elements in plants [37,44–47].

Additionally, the growth parameters presented a positive response for nozzles utilizing air compared to nozzles not utilizing air, while little growth was observed for the ultrasonic nozzle misting at 30-, 45- and 60-min intervals. Continuous contact with oxygen for air-assisted atomizers stimulates metabolic processes, which has positive effects on the development of shoots, roots and nutrient uptake [48]. The growth trend of all parameters was different between all treatments. The number of leaves was not significantly different between N2, N3 and N4 misting at 30-, 45- and 60-min intervals. The highest stem diameter, leaf length, leaf with and leaf area were observed for N1 compared to those observed for N2, N3 and N4 at the same spraying intervals, and the lowest above-mentioned parameters were found for the ultrasonic nozzle. The number of leaves, stem diameter, leaf length, leaf width and leaf area showed decreasing trends under N1 and N2 with increasing spraying interval; however, these parameters first increased and then decreased with 30-, 45- and 60-min spraying intervals. The findings of our research showed a very close connection with the review literature that the droplet size and spraying interval had powerful relations to plant growth [20,22,26,30,49–51]. In another study, the authors of Reference [21] reported that plant growth could be improved by using proper droplet size and that air-assisted atomizers presented higher yields than those of airless nozzles.

Furthermore, the biomass yield and edible yield results showed a decreasing trend for all treatments with increasing spraying interval. The results showed that spraying intervals had an inversely proportional relationship to biomass yield and edible yield. Significantly

higher biomass yield and edible yield were observed for the nozzle with air (N1) than those observed for N2 and N3 misting at a 30-min interval. Very poor plant yield was observed for the ultrasonic nozzle. This was because of the high change in pH and EC values of the nutrient solution. These results agree with the findings of Lakhiar et al. [38] that a relatively high biomass yield could be obtained from lettuce with an optimal nutrient solution recycling process. Li et al. [52] also reported that aeroponic systems with proper spraying intervals lead to remarkably improved biomass yield.

Regarding the two-way ANOVA results, the root characteristics were more dependent on droplet size and spraying interval than the leaf characteristics. However, N1 showed a significant ($p < 0.05$) effect on all measured parameters. N3 showed a mixed phenomenon of a significant ($p < 0.05$) and nonsignificant ($p > 0.05$) effect, and N2 and N4 presented a nonsignificant ($p > 0.05$) effect on root characteristics. A greater root length, root diameter, root area and root volume were observed with the air-assisted nozzle and a 30-min spraying interval than with N2, N3 and N4 misting at 30-, 45- and 60-min intervals. It is well known that a relatively large root system could lead to a higher biomass yield than a small root system. Previous studies also present the same type of results: good aeration of the root environment is most advantageous for root growth [53]. Lakhiar et al. [21] also reported similar results, that air-assisted atomizers enhance root growth more significantly ($p < 0.05$) than airless atomizers.

The analyzed results indicated that the droplet size (nozzle) and spraying interval affected the root-to-shoot ratio (fresh and dry). An increase in the fresh root-to-shoot ratio with increasing misting interval was observed for N1, N2 and N4, while for N3, the root-to-shoot ratio first increased and then decreased at 30-, 45- and 60-min spraying intervals. Furthermore, an increasing in the dry root-to-shoot ratio was calculated for N1 and N4 with increasing spraying intervals, and a decrease in this parameter was observed for N2 and N3. The researchers [53–55] also reported corresponding findings that the root-to-shoot ratio was higher in aerated growth boxes with good oxygen circulation than under poor oxygen conditions. Another study concluded that different spraying intervals induce different integrated distribution patterns, shifting assimilates from roots to shoots [56–59].

The interaction effect of droplet size and nutrient solution spraying intervals showed a positive effect on the nutrient uptake of aeroponically grown lettuce. It was observed that the nozzle with air (N1) had a significant ($p < 0.05$) effect on phosphorus, potassium and magnesium uptake and a nonsignificant ($p > 0.5$) effect on nitrogen and calcium. N3 showed a significant ($p < 0.05$) effect only on magnesium uptake, and the ultrasonic nozzle presented a significant ($p < 0.05$) effect on calcium uptake. More importantly, N2 illustrated a nonsignificant ($p > 0.05$) effect on all measured parameters of nutrient uptake. These elements benefit plants directly or indirectly by affecting plant metabolism. The findings of our research coincide with the recommendations of Khan et al. [33], Li et al. [52] and Xie et al. [60], who reported that N, P, K, Ca and Mg uptake increases the efficiency of photosynthesis, which is key to increasing crop yield in an aeroponic system. Another study reported that good aeration with proper spraying intervals of the root environment is advantageous in aeroponics and could result in improved uptake of N, P, K, Ca and Mg by plants [61,62].

## 5. Conclusions

This study demonstrated that both droplet size (nozzle) and nutrient solution spraying interval are major factors affecting shoot and root growth of lettuce crops grown in aeroponic systems. The biomass yield was significantly higher under the N1I1 treatment than under the N2, N3 and N4 treatments misted at 30-, 45- and 60-min intervals. For the air-assisted nozzle misting at a 30-min spraying interval, shoot development was more constrained than root development, which was prominent in the alteration of the root-to-shoot ratio (fresh and dry) and nutrient uptake. The authors concluded that the nozzle utilizing air (N1) provides a suitable environment, oxygen availability and is more advantageous for root growth. The greater root growth provided greater shoot biomass (yield) as compared

to airless and ultrasonic nozzles misted at different spraying intervals. In addition, to grow crops and vegetables in aeroponic systems, we recommended that the proper droplet size, spraying interval and nutrient solution for each cultivar be investigated in the future.

**Author Contributions:** M.H.T. collected the data, designed the experiment, involved in laboratory analyses and drafted the manuscript. J.G. arranged all tests, designed all nozzles and gave main suggestions how to analyze the data, edited and finalizes the manuscript. I.A.L. gave some important suggestions for the initial manuscript. W.A.Q. was also involved in laboratory analyzes. S.A.S. and K.A.S. helped in data collection and removed the mistakes and J.C. finalized the manuscript. All authors have read and agreed to the published version of the manuscript.

**Funding:** We acknowledge that this work was financially supported by the National Natural Science Foundation of China Program (No. 51975255), Jiangsu Agriculture Science and Technology Innovation Fund (CX (18) 3048), and the Project Funded by the Priority Academic Program Development of Jiangsu Higher Education Institutions (No. PAPD-2018-87).

**Data Availability Statement:** The data is available in Figures 5–11 and Tables 1–3 within the article.

**Conflicts of Interest:** The authors declare no conflict of interest.

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
