# Peer review of "Influence of Atomization Nozzles and Spraying Intervals on Growth, Biomass Yield, and Nutrient Uptake of Butter-Head Lettuce under Aeroponics System"

_agronomy, doi:10.3390/agronomy11010097_

Round 1

Reviewer 1 Report

In my opinion, the experimental design is a split-plot and the data must be treated and presented under this criterion. The authors do not show if there are significant differences between treatments and do they venture to show trends directly. In a general way for all parameters.
I think they should rewrite the results part with this consideration. I think it is a more effective way of understanding what happens.

Regarding aspects of format to the treatments, they should be sub-indexed, for example, N1 .

On the other hand, it might be convenient to order the treatments N by the nozzle size, since currently, N1 is 11 N2 is 26 N3 is 17 and N4 is 4.89, this makes it difficult for the reader to understand the results.

In material and methods, the nutritive solution must be put, not the fertilizer salts used for this purpose.

I would also recommend for future trials. that the authors use between 60 and 80 oC instead of 105 oC to dry the plant material, since N losses due to volatilization increase 

Author Response

Manuscript ID- Agronomy-1011867

Influence of Atomization Nozzles and Spraying Intervals on Growth, Biomass Yield, and Nutrient Uptake of Butter-head Lettuce under Soilless Cultivation System

Dear Anonymous Reviewer,

We would like to thank the referee for reviewing, and providing many useful suggestions to make our manuscript more effective. We revised the manuscript and tried to our best by incorporating the suggestions of reviewers properly. In this response, we are presenting a point-by-point summary of the reviewer's comments.

Reviewer 1: Comments and Suggestions for Authors

Comment 1: In my opinion, the experimental design is a split-plot and the data must be treated and presented under this criterion. The authors do not show if there are significant differences between treatments and do they venture to show trends directly. In a general way for all parameters.

Response: Thank you very much for your valuable suggestions. The aeroponics systems were arranged in a randomized complete block design (RCBD) with 12 treatments. The experiment was comprised of four atomizing nozzles (one with air, two without air and one ultrasonic nozzle) and three nutrient solution spraying intervals (30, 45 and 60 minutes). We have improved our results and the trend line clearly shows the significant different in each treatment. We have improved our graphs also and inserted error bars and ANOVA significance difference.

Comment 2: I think they should rewrite the results part with this consideration. I think it is a more effective way of understanding what happens.

Response: It is also a good suggestion; we have already presented description of each parameter very clearly.

Comment 3: Regarding aspects of format to the treatments, they should be sub-indexed, for example, N1. On the other hand, it might be convenient to order the treatments N by the nozzle size, since currently, N1 is 11 N2 is 26 N3 is 17 and N4 is 4.89, this makes it difficult for the reader to understand the results.

Response: Yes off course, the values presented by the anonymous reviewer are absolutely correct, for your kind information that we have presented results on the basis of nozzles type not on droplet size and according to the design of systems consist of different type of nozzles. We have presented the droplet size in the result part for the ease of understand. 

Comment 4: In material and methods, the nutritive solution must be put, not the fertilizer salts used for this purpose.

Response: Thank you very much for your valuable suggestion, we have added nutrient solution used during the experiment.

Comment 5: I would also recommend for future trials, that the authors use between 60 and 80 oC instead of 105 oC to dry the plant material, since N losses due to volatilization increase 

Response: Thank you very much! In future trials we will take care of this thing.

Reviewer 2 Report

In this form the manuscript cannot be accepted: it contains many gaps. I point out some of them below.
The keywords n. 6 and 7 are contained in the title of the article.
Bibliographic references 1, 2, 3, 6, 11, 25, 43 (like others) are not pertinent to what is reported (it is necessary to refer to specific papers of the statement made).
The sentence given in lines 50-51 is not supported by bibliographic reference n. 19.
The bibliographic reference n. 26 is not helpful in proving the statement in lines 54 and 55.
The sentence on lines 57-59 is not understandable.
The phrase on lines 68-69 is also not well written.
What scientific evidence is there to support the statements in lines 71-74?
All lines 81-82 are attributed four references for a claim that does not need experimental scientific evidence (they are not relevant).
Line 86: delete "significant".
How are the experimental N2 and N3 treatments different?
How to get 192 plants in total (line 144).
How many liters of nutrient solution were used for each elementary experimental unit?
Line 147: (NH4)6Mo7O24 insert the subscripts correctly.
How EC and pH were maintained (line 149).
How long did the crop cycle last?
Line 161: here are four bibliographic references for a series of determinations which are then described (these bibliographical references are useless).
Which experimental treatments do the photographs in figure 4 refer to?
Paragraph 2.8: on what have the mineral elements been determined?
The reference n. 49 is relevant (line 187)?
The manuscript does not show the results of the ANOVA and is limited to a presentation of the figures in which the trends of dependent variables connected to three levels of the experimental treatment are reported (from three points it is very likely that a straight line passes; from three points there always passes a parable ...). It would be better if you developed the variance analysis with the study of polynomial contrasts.
The statement reported on lines 216-217 is incorrect: with N4 the trend is to decrease.
What do ΔEC and ΔpH mean?
Figure 6: With N2 the trend shown in the figure is quadratic, while the equation shown is that of a straight line. A parable always passes from three points ...
In the figures it is not clear which treatments the equations apply to.
In the figures it is necessary to report the standard error of the means (and results of ANOVA).
Several bibliographic references are incomplete (for example, n. 17, 24, 28, 32 are book chapters? What does n. 20 refer to? n. 26 is inappropriate.
It is necessary to check all the bibliography (for example n. 22).

Author Response

Manuscript ID- Agronomy-1011867

Influence of Atomization Nozzles and Spraying Intervals on Growth, Biomass Yield, and Nutrient Uptake of Butter-head Lettuce under Soilless Cultivation System

Dear Anonymous Reviewer,

We would like to thank the referee for reviewing, and providing many useful suggestions to make our manuscript more effective. We revised the manuscript and tried to our best by incorporating the suggestions of reviewers properly. In this response, we are presenting a point-by-point summary of the reviewer's comments.

Reviewer 2: Comments and Suggestions for Authors

In this form the manuscript cannot be accepted: it contains many gaps. I point out some of them below.

Comment 1: The keywords n. 6 and 7 are contained in the title of the article.
Response: We have changed the mentioned keywords due to the presence in tittle.  

Comment 2: Bibliographic references 1, 2, 3, 6, 11, 25, 43 (like others) are not pertinent to what is reported (it is necessary to refer to specific papers of the statement made).

Response: We have much improved the paragraphs including inappropriate bibliographic references. Now, it is very clear to observe from the reported sentences.

Comment 3: The sentence given in lines 50-51 is not supported by bibliographic reference n. 19.

Response: Thank you very much for you valuable suggestion, the authors have improved the lines. 

Comment 4: The bibliographic reference n. 26 is not helpful in proving the statement in lines 54 & 55.

Response: We have improved the mentioned statements. 

Comment 5: The sentence on lines 57-59 is not understandable.

Response: We have tried to our best to make sentence easy to understand.

Comment 6: The phrase on lines 68-69 is also not well written.

Response: The authors have rewritten the phrases written in the lines 68-69.

Comment 7: What scientific evidence is there to support the statements in lines 71-74?

Response:  This statement supported by the researcher Buckseth et al. (2016).

Comment 8: All lines 81-82 are attributed four references for a claim that does not need experimental scientific evidence (they are not relevant).

Response: Thank you very much! For your valuable suggestion, we have much improved and deleted the inappropriate references.

Comment 9: Line 86: delete "significant".

Response: We have deleted the word significant

Comment 10: How are the experimental N2 and N3 treatments different?

Response: The experimental treatments including N2 and N3 nozzles are different on the basis of average droplet sizes. It is therefore the N2 had droplet size of 26.35 µm and N3 had average droplet size of 17.38 µm.

Comment 11: How to get 192 plants in total (line 144).

Response:  Sorry, by mistake we wrote incorrect plants number; we have corrected the number of plants in the manuscript.

Comment 12: How many liters of nutrient solution were used for each elementary experimental unit?

Response: We have already provided the detail of the nutrient supply in through each aeroponic system. It could be seen from the schematic diagram that only one air-assisted nozzle/atomizer, four air-less and four ultrasonic nozzles are consisted in the systems. Hence, for the balancing and equal distribution of nutrient flow rate; the air-assisted atomizer (N1) misted 4 liter per minute, each airless (N2 & N3) and ultrasonic nozzle (N4) misted 1 liter per minute throughout the experiment.   

Comment 13: Line 147: (NH4)6Mo7O24 insert the subscripts correctly

Response: Thank you very much for indication we have corrected as (NH4)6Mo7O24•4H2O.

Comment 14: How EC and pH were maintained (line 149).

Response: The procedure followed for maintaining the pH and EC was (Singh, P and Bruce, D., 2016; Yang et al., 2020).

Comment 15: How long did the crop cycle last?

Response: The crop harvest period was 40 days after transplant (DAT).

Comment 16: Line 161: here are four bibliographic references for a series of determinations which are then described (these bibliographical references are useless).

Response: Yes, the authors also realize after reading that the references in line 161 were useless. It is therefore we have much improved the lines.

Comment 17: Which experimental treatments do the photographs in figure 4 refer to?

Response: The root growth in the figure four is journal in all treatments, we only shows the root pattern during the experiment. If it is necessary we can insert treatment wise.  

Comment 18: Paragraph 2.8: on what have the mineral elements been determined?

Response: Thank you for suggestion to improve the paragraph. We have tried our best to present paragraph more easy to understand. 

Comment 19: The reference n. 49 is relevant (line 187)?

Response: We have deleted the reference, it was irrelevant. 

Comment 20: The manuscript does not show the results of the ANOVA and is limited to a presentation of the figures in which the trends of dependent variables connected to three levels of the experimental treatment are reported (from three points it is very likely that a straight line passes; from three points there always passes a parable ...). It would be better if you developed the variance analysis with the study of polynomial contrasts.

Response: We have much improved the graphs, and we have presented the ANOVA results in the descriptions written related to graph. If the anonymous reviewer suggest, the author will change the graph type.

Comment 21: The statement reported on lines 216-217 is incorrect: with N4 the trend is to decrease

Response: We are extremely sorry for incorrect statement, the anonymous reviewer is absolutely right. We have corrected the sentences.

Comment 22: What do ΔEC and ΔpH mean?

Response:  It means that change in pH and change in EC values. We have also corrected in the manuscript

Comment 23: Figure 6: With N2 the trend shown in the figure is quadratic, while the equation shown is that of a straight line. A parable always passes from three points ...

Response:  We have corrected not only the N2 trend line but also in other graphs, where it was necessary to insert the quadratic line. 

Comment 24: In the figures it is not clear which treatments the equations apply to.

Response: We have mansions in the graphs and inserted equations in the blank spaces to understand easily, whereas in the models yn = Nn

Comment 25: In the figures it is necessary to report the standard error of the means (and results of ANOVA).

Response: We have inserted the standard error of means in all graphs.

Comment 26: Several bibliographic references are incomplete (for example, n. 17, 24, 28, 32 are book chapters?

Response: We have corrected the bibliographic references and deleted the inappropriate references.

Comment 27: What does n. 20 refer to? n. 26 is inappropriate.

Response:  The authors have also deleted the unnecessary references.

Comment 28: It is necessary to check all the bibliography (for example n. 22).

Response: We have checked all references and some of them improved, however, unnecessary and inappropriate bibliographic references deleted.  

Reviewer 3 Report

This paper presents and interesting topic such as the “Influence of atomization nozzles and Spraying intervals on growth, biomass yield and nutrient uptake in a soilless cultivation system”.

The paper presents a detailed work and a suitable experimental design with different treatments and repetitions to be able to evaluate these effects. Experimental design such as data processing, discussion and conclusions can be considered correct.

Nevertheless, there are some points that need to be clarified or extended before its publication:

The authors need to do a much more complete review of the state of the art, incorporating references more focused on the points they specify such as the influence of climate change on production, the importance of water scarcity, the importance of local production and city production; the need to reduce water and fertilizer consumption, etc.

It seems that the literature review should be increased prior to the publication in order to get a better job and to help increase the quality of the article

Other small changes to consider:

Page 1 line 39 “have been  significant” please remove the double space

Page 4 line 123 “Flow rate of L.min-1” The point must be deleted.

Page 4 line 149 you need to change “dS m-1”to dS m-1

Page 4 line 148-149  please explain this sentence better.

Page 14 line 369-370 Regarding the expression “Additionally, the growth parameters presented a positive response for nozzles utilizing air compared to nozzles not utilizing air,” can the authors suggest any idea of why this happens? Any references to comment on it? some suggestion from the authors. Perhaps due to greater oxygenation of the nutrient solution of the drops?.

Later the authors in Page 15 line 399-400 introduce this idea “Previous studies also present the same type of results: good aeration of the root environment is most advantageous for root growth” Can authors cite  any of the studies? Can the authors better link the conclusions on these two points?

Finally a suggestion:

IF the study is done in all cases in aeroponics would it not be better to modify the title and remove soilless and include aeroponic?

Author Response

Manuscript ID- Agronomy-1011867

Influence of Atomization Nozzles and Spraying Intervals on Growth, Biomass Yield, and Nutrient Uptake of Butter-head Lettuce under Aeroponics Systems

Dear Anonymous Reviewer,

We would like to thank for reviewing, and providing many useful suggestions to make our manuscript more effective. We revised the manuscript and tried to our best by incorporating the suggestions properly. In this response, we are presenting a point-by-point summary of comments.

Reviewer 3: Comments and Suggestions for Authors

This paper presents and interesting topic such as the “Influence of atomization nozzles and Spraying intervals on growth, biomass yield and nutrient uptake in a soilless cultivation system”.

The paper presents a detailed work and a suitable experimental design with different treatments and repetitions to be able to evaluate these effects. Experimental design such as data processing, discussion and conclusions can be considered correct.

Nevertheless, there are some points that need to be clarified or extended before its publication:

Comment 1: The authors need to do a much more complete review of the state of the art, incorporating references more focused on the points they specify such as the influence of climate change on production, the importance of water scarcity, the importance of local production and city production; the need to reduce water and fertilizer consumption, etc.

Response: We have much improved the state of art and complete the review by adding more references and added these lines: Through flooding, hurricane, storms, and droughts have drastically reduced agriculture land (Touliatos et al., 2016; Muller et al., 2017).  Scientists predicted that adverse weather conditions and climate change will result in the deprivation of the large areas of arable land, rendering them unstable for farming (Padmavathy et al., 2016; Thomaier et al., 2015).

Comment 2: It seems that the literature review should be increased prior to the publication in order to get a better job and to help increase the quality of the article

Response: Thank you very much for your valuable suggestion. We have increased some literature review and it reached at 63 references.    

Other small changes to consider:

Comment 3: Page 1 line 39 “have been significant” please remove the double space

Response: We have removed the double space

Comment 4: Page 4 line 123 “Flow rate of L.min-1” The point must be deleted.

Response: The authors have deleted the point in between acronym

Comment 5: Page 4 line 149 you need to change “dS m-1”to dS m-1

Response: Yes, we have changed into superscript

Comment 6: Page 4 line 148-149 please explain this sentence better.

Response: Thank you very much! We have tried better to explain the sentence 

Comment 7: Page 14 line 369-370 Regarding the expression “Additionally, the growth parameters presented a positive response for nozzles utilizing air compared to nozzles not utilizing air,” can the authors suggest any idea of why this happens? Any references to comment on it? some suggestion from the authors. Perhaps due to greater oxygenation of the nutrient solution of the drops?

Response: The authors suggested that continuous contact with oxygen for air-assisted atomizers stimulates metabolic processes, which has positive effects on the development of shoots, roots and nutrient uptake (Stoner, R.; and Clawson, J., 1998).

Comment 8: Later the authors in Page 15 line 399-400 introduce this idea “Previous studies also present the same type of results: good aeration of the root environment is most advantageous for root growth” Can authors cite any of the studies? Can the authors better link the conclusions on these two points?

Response: We have inserted the reference in the mentioned place. The previous studied such as (Clayton and Lamberton 1964; Cho et al. 1996; Park and Chiang 1997; Burgess et al. 1998; Garrido et al. 1998a; Garrido et al. 1998b; Scoggins and Mills 1998; Molitor et al. 1999; Kamies et al. 2010) also supports our study.  We have also linked these two points with conclusions by adding lines” The authors concluded that the nozzle utilizing air (N1) provides a suitable environment, oxygen availability, and more advantageous for root growth. The greater root growth provided greater shoot biomass (yield) as compared to air-less and ultrasonic nozzles misted at different spraying intervals”.   

Finally a suggestion:

Comment 9: IF the study is done in all cases in aeroponics would it not be better to modify the title and remove soilless and include aeroponic?

Response: Thank you very much; we have replaced soilless cultivation system with aeroponics systems

Round 2

Reviewer 1 Report

The authors have answered the required questions.

Author Response

Manuscript ID- Agronomy-1011867

Influence of Atomization Nozzles and Spraying Intervals on Growth, Biomass Yield, and Nutrient Uptake of Butter-head Lettuce under Aeroponics Systems

Dear Anonymous Reviewer,

The authors are really thankful to you for your valuable and positive comment.   
